# Risk factors for early childhood disability in Bangladesh: Evidence from Multiple Indicator Cluster Survey 2019

**Shilpi Rani Saha** [ID][1]*, **Md. Mobarak Hossain Khan**[2]

**1** Bangladesh University of Professionals, Dhaka, Bangladesh, **2** Department of Social Relations, East West University, Dhaka, Bangladesh

* ssashilpi@gmail.com

**Data Availability Statement:** The data underlying the results presented in the study are available from https://mics.unicef.org/surveys.

**Funding:** The authors received no specific funding for this work.

## Abstract

### Introduction

Early childhood is a vital part of human life because most of the brain developments occur in this particular period. Early childhood disability is a significant global public health burden, which can negatively impact the children's quality of life and their overall productivity. It is also a major social and economic problem in Bangladesh. Therefore, it is very important to understand the associated factors for early childhood disability, which may help disability prevention, better management and policy formulation. The main objective of this study is to investigate the child, family, and community-level factors associated with early childhood disability in Bangladesh.

### Methods

A cross sectional nationally representative data was derived from Multiple Indicator Cluster Survey (MICS), 2019. A total of 14,072 Bangladeshi children under five years of age were selected for this study. Various types of statistical analysis (simple, bivariate, multivariable) were performed. To assess the bivariate relationship between chosen categorical variables (independent) and early childhood disability (dependent), a chi-square test was used. The multivariable ordinal logistic regression was used to find out the association of disability with child, family, and community-level factors.

### Results

The results show that 2.0% of the children have at least one disability and 0.8% have more disabilities. Several factors namely not attending in early childhood education [Odds Ratio (OR) = 0.65; 95% confidence interval (CI) = 0.13–1.17 P = 0.01], having mother's functional difficulty (OR = 1.23; 95% (CI) = 0.58–1.88 P <0.001), unhappy mother's life (OR = 0.85; 95% CI = 0.30–1.39 P <0.001), parents without internet access (OR = 0.68; 95% CI = 0.06–1.29 P = 0.03) and parents using mobile phone (OR = 0.52; 95% CI = 0.09–0.95 P = 0.02) were found to be important for early childhood disability in Bangladesh.

**Competing interests:** The authors have declared that no competing interests exists.

## Conclusion

Early childhood disability is still neglected in Bangladesh and further epidemiological studies are recommended. The findings of this study may help policy makers and relevant stake-holders to develop interventions for reducing the overall burden of early childhood disability.

## Introduction

The first five years of life are generally known as early childhood. This is a vital part of life because a child's major growth and development, especially brain development occur in this particular period [1]. Although under-five child mortality is reducing in low and middle-income countries, early childhood disability is still on the rise [2]. Approximately 200 million under-five children in developing countries fail to reach their potential development because of poverty, malnutrition, and poor health [3]. Vision loss, hearing loss, neonatal diseases, pre-term birth, and infections were the leading causes for under five developmental disabilities [1].

Without achieving the full potential of disabled children, it could be difficult to achieve the sustainable goals (SDGs) referring to quality education (Goal 4), no poverty (Goal 1), zero hunger (Goal 2) and good health and wellbeing (Goal 3) for all children including those with disability by 2030 [1].

Reliable definitions are still insufficient to measure childhood disability [4]. Generally, children who are unable to lead a normal life due to physical and mental defects are called disabled and this condition can limit human functioning in multiple ways [5]. It is associated with illness, disorder, or other health-related conditions, restriction of participation, and activity limitation in the context of one's environment [5]. Disability can be categorized into two main dimensions called physical disability and learning disability. Sometimes, it may be the result of the overlapping of two dimensions [6]. Briefly, physical disability is a condition that mainly reduced individual mobility, self-care and usual life activities [6]. Similarly, learning disability is a condition that cause people to take longer time to learn and may require assistance to develop new skills and interacting with others [6]. Physical health may decline rapidly and early death can occur due to childhood disability [7]. Worldwide, disable children are increasing as a global burden [8]. Childhood disability not only limiting the children's physical, mental, social, and economic conditions but also affects his/her families and societies [6]. To provide necessary support and services to the disable children, their families give considerable time and efforts which tend them to experience economic burden, job loss, poor psychological and physical health, and negative social consequences [9]. Also, children born in poor-income countries have a higher risk of disability because of poor ante-natal care, poor health-related services and, poverty [3].

The prevalence of developmental disability among under-five children varied from region to region (Fig 1) [1]. In 2016, globally 52.9 million children under five years of age had a developmental disability and the prevalence rate was 8.4%. Among them, 95% were living the low- and middle-income countries [1]. The highest number of developmental disability cases among under-five children was reported in the South Asia region (15.0 million) and the lowest in the Oceania region (0.11 million). Sub-Saharan Africa were also severely affected by this problem [1].

Although early childhood disability is increasing in Bangladesh, unfortunately reliable data on the disability-related issues is still limited [10]. Moreover, the prevalence rate is also under-estimated and varied due to inadequate definition of disability and lack of systematic statistical

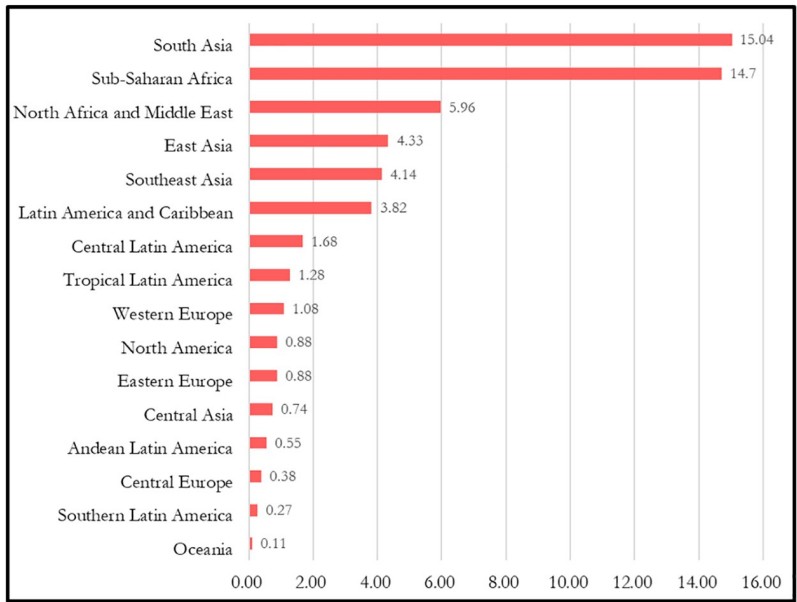

**Fig 1. Prevalence (cases) of under-five developmental disability by region.** (Source: [1]).

data collection process [10]. According to the 2011 Population and Housing Census data [11], under-five disability rate was 0.5%. Among them 0.49% had some sort of disability, 0.3% had severe disability, and 0.14% were fully disabled. Olusanya et al. (2018) [1] reported that under-five children with developmental disability was 1.3 million in Bangladesh in 2016.

Three general conceptual models have been adopted to better understand the disability [6]. The model includes medical (individual physical problem), social (product of environment), and the interaction between medical and social aspects [6]. According to the medical model, disability is associated with pathology. It could be cured by medicine or healthcare professionals [6]. Whereas the social model describes the disability in such a way that, disability is not an individual problem, it is a failure of the society not to provide appropriate support and services to disabled persons [6]. So, it can be argued that disability is not the medical concept nor the social concept. It is a combination of medical and social models [12]. The combined idea of the medical and social model could be referred to as the biopsychosocial model [12]. George Engel first conceptualized the Biopsychosocial model in 1977 [12]. Based on this combination, the WHO [13] developed the International Classification of Functioning, commonly known as ICF framework for describing disability (see Fig 2). The ICF provides a clear view of the different health perspectives: biological, individual, and social [14].

According to the ICF model, the word 'functioning' applies to all body functions and structures, activities, and participation, while disability refers to impairments, activity limitations, and participation limits [15].

The ICF model for Children and Youth Version (ICF-CY) was issued by WHO in 2007 [13]. To conceptualize disability, it provides three important wide-ranging components [10]. According to ICF-CY, disability is the inability to visualize shape or size, activity limitation, such as inability to move around and to read text, participation restriction, such as elimination from school [10]. So, the term 'children with disability' include a vast range of activity limitation and severity of disability [10].

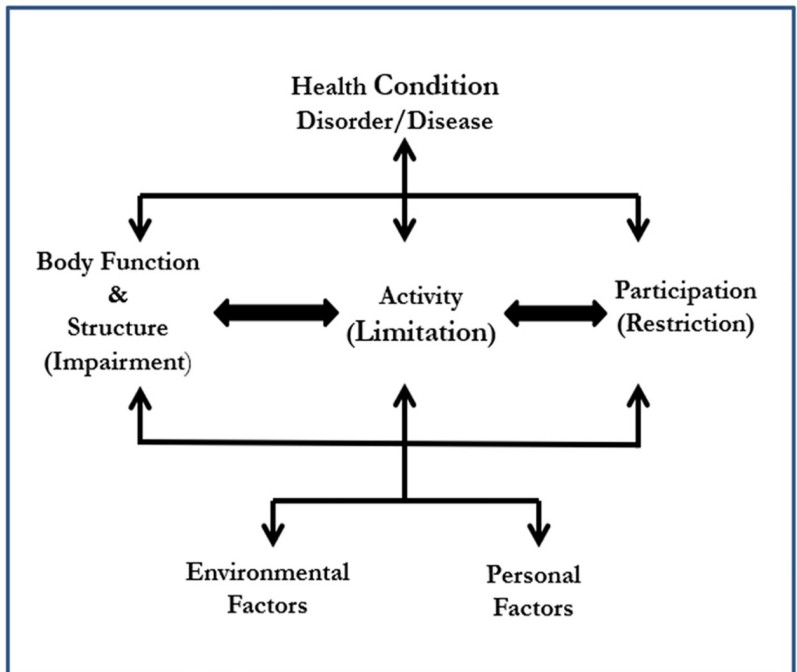

**Fig 2. The ICF model for measuring health and disability (Source: [13]).**

Different child, family, and community-level factors are associated with early childhood disability. Child-level factors include low birth weight, genetic diseases [2], and neurodevelopmental problems [16]. Among the family-level factors, the mother with older age, mother illiteracy [2], mothers psychological wellbeing including maternal depression, intimate partner violence and drug use [16], poor socioeconomic conditions of the families [4, 16] are important. Community-level factors such as children's delivery at home, and lack of antenatal care [2] are also strongly associated with childhood disability. The prevention and management of disability depends on the reliable epidemiological data (e.g., distribution of burden and causes) [17].

Durkin et al. (2000) [18] explored the two-phase national household survey data (1987, 1988) and found that maternal goiter, postnatal brain infections, maternal illiteracy, landlessness, maternal history of pregnancy loss, and gestational age at birth are the significant risk factors for childhood disability in both urban and rural areas of Bangladesh [18]. Another study conducted by Jahan et al. (2019) [19] found that malnutrition is one of the major causes of early childhood disability in Bangladesh. In rural Bangladesh, infectious diseases tend to be one of the leading causes of serious childhood disability [20]. Few studies related to adult disability reported prevalence and risk factors [21–23]. These studies identified poverty, older age, sex, lower educational level, marital status, place of residence, smoking, and diabetics with higher blood pressure as the risk factors for (adult) disability [21–23]. Environmental factors including chemicals, human-made changes to the environment, lead exposure, cigarette smoke, industrial air pollution, and toxins in the air are also found as significant factors for adult disability [15, 16].

Childhood disability is a significant burden for both low- and middle-income countries including Bangladesh [2]. Moreover, in Bangladesh, there is a common myth that children

with disability is a social stigma and is considered as a punishment for parents' sinful actions [10]. Disable children are deprived of getting education and healthcare services. Sometimes they are the most vulnerable to violence, abuse, exploitation, and neglect when they are placed in institutions [10]. Gender is also a contributing factor, as disabled girls are less likely than boys to receive food and care [10].

Disability related data. particularly, reliable data on causes and factors for early childhood disability in Bangladesh are scare [20]. According to our knowledge, a well-documented study on factors affecting early childhood disability is still limited in Bangladesh [18–22]. Previous research was unable to find the basic relationship between early childhood disability and associated factors. These studies may suffer from various limitations, such as inadequate focus on specific (cognitive) disability [18], use of limited statistical methods [19], and small sample size with the specific region [20]. Although, some of the studies used multivariable linear and binary logistic regressions to identify the potential determinants of childhood disability [18, 21, 22], none of the studies used ordinal logistic regression to find out the child, family, and community-level determinants.

Identifying both risk factors and the impact of disability is an important process to limit or prevent disability [20]. Additionally, early intervention may reduce the impact of disability. Without a proper understanding of the burden of childhood disability, it could be difficult to provide them adequate services. Social support and early intervention could help them to reach their full potential as SDGs mandate [8], such as to ensure inclusive and equitable quality education and promote lifelong learning opportunities for all children [24].

Therefore, this study aims to report some factors that may affect early childhood disability in Bangladesh using secondary data. Particularly, child, family, and community-level factors are investigated using ordinal logistic regression. The findings of this study can be used for policy formulation to provide adequate support and services for children with disabilities. These findings may also suggest some strategies to prevent disability and early diagnosis and intervention.

## Materials and methods

### Study design

The secondary data from the latest round of Multiple Indicator Cluster Survey (MICS) 2019 was used for this study. The MICS 2019 was conducted by the Bangladesh Bureau of Statistics (BBS) and the Ministry of Planning from January 19 to June 1, 2019 [25]. UNICEF, Bangladesh provided both technical and financial support to conduct this survey. This survey collected data for 144 major indicators, of which 29 indicators are directly related to the Sustainable Development Goals (SDGs). Since detailed information about the survey methodology is available in the published report (BBS and UNICEF Bangladesh, 2019) [25], only some useful information is provided below.

The MICS was a cross-sectional household survey based on two-stage stratified cluster sampling technique. This survey produced nationally representative statistically sound and internationally comparable data. The MICS data is extensively used for national level policymaking and interventions. Five types of questionnaires were used for data collection during the survey. These were (i) a household questionnaire; (ii) a water quality testing questionnaire, (iii) a questionnaire for individual women aged 15–49 years, (iv) a questionnaire for children under-5 and (v) a questionnaire for children aged 5–17 years. Mothers or caregivers were asked to collect data while administering the questionnaire surveys for children under-five and 5–17 years (see report BBS and UNICEF Bangladesh, 2019) [25].

## Sampling techniques

A two-stage stratified cluster sampling approach was employed to select the survey sample. The sampling frame was based on the 2011 Bangladesh Census of Population and Housing. In total, there were 293,533 census enumeration areas (urban = 65,193, rural = 228,340) in the whole country, divided into 8 divisions (Barishal, Chattogram, Dhaka, Khulna, Mymensingh, Rajshahi, Rangpur, and Sylhet) and 64 districts. At the first stage, a total of 3,220 primary sampling units (PSUs) were systematically selected from these enumeration areas (EAs). At the second stage, a listing of households was conducted in each of the selected EA. Then from these households, a systematic sample of 20 households per PSU was drawn for data collection (see Fig 3). The total number of sampled households were 64,000, which was determined by using well-defined statistical formula. The detailed methodology of survey including sample size calculation is available in the report (BBS and UNICEF Bangladesh, 2019) [25].

From 64,000 number of sampled households, a total of 61,242 households were successfully interviewed. Moreover, the survey gathered information for 80,776 children aged 2 to 17 years. Out of them, only 14,072 children were under five years of age group [25]. The whole analysis was performed based on these under five children (target sample).

## Measuring childhood disability and outcome variables

Since this study focused on childhood disability, some more information is given about child functioning including childhood disability. According to The United Nations Convention on the Rights of Persons with Disabilities (CRPD) and the International Classification of Functioning for Children and Youth (ICF-CY), childhood disability can be assessed through the different major functioning domains. The child functioning domain for children under-5 (from 2–4 years of age) includes: seeing, hearing, walking, fine motor, communication, learning, playing, and controlling behavior. The disability level is estimated by asking whether the child has no difficulty (coded as = 1), or some difficulty (= 2), or a lot of difficulty (= 3), or cannot do it at all (= 4). For controlling behavior, how often the child kick, bite or hit other children compared with children of the same age (1 = not at all, 2 = less, 3 = the same, 4 = more, 5 = a lot more). Functional difficulty in the individual domain (except controlling behavior) was calculated if the mother or primary caregivers reported that the child had a lot of difficulties (code = 3) or cannot do it all (code = 4) and a lot more (code = 5) for controlling behavior. If the mother or primary caregivers reported that a child has any functional difficulty they were characterized as screening positive [25].

## Independent variables

Child-level, family-level, and community-level variables were considered to find out their association with childhood disability. Child-level variables were child age (2 to 4 years), child sex (male, female), early childhood education attendance (attending, not attending), measures of underweight (not underweight, underweight), measure of stunting (not stunted, stunted), birth order (first born, second born, third up to last). Family-level variables included mothers age (15–19 years, 20–24 years, 25–29 years, 30–34 years, 35–39 years, 40–44 years, and 45–49 years), mother's education (pre-primary or none, primary, secondary and higher secondary+), mother's functional difficulty (has functional difficulty, has no functional difficulty and no information), mother's age at birth (<20, 20–34, 35+) and mother's overall happiness (very happy, somewhat happy, neither happy nor unhappy, somewhat unhappy, very unhappy). Community-level variables were the place of residence (urban, rural), antenatal care visit to any provider during her last pregnancy as recommended by WHO (1 to 4 visits, 5+ visits), delivery at home (yes, no), ever use of internet (yes, no) and own a mobile phone (yes, no).

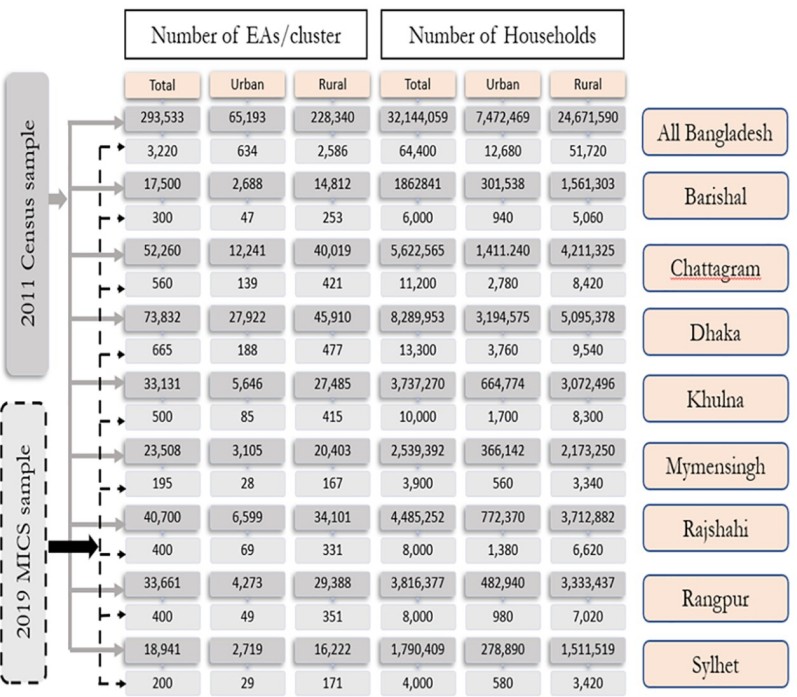

**Fig 3. Sample size and sample allocation (Source: [25]).**

## Measurement of underweight and stunting

Underweight is defined as a weight that is 2 standard deviations below the WHO child growth standards for that particular age. In other words, the child is underweighted if Z-scores of children for a given weight for age is less than -2 SD from the median of the WHO Child Growth Standards or References. Similarly, stunting is defined as a height that is more than 2 standard deviations below the WHO child growth standards median. In other words, the child is stunted if Z-scores of children for a given height for age is less than -2 SD from the median of the WHO Child Growth Standards or References [25].

## Statistical analysis

To conduct childhood disability analysis, we needed to merge the dataset. As the first stage of analysis, we used childhood disability-related factors including SDG-related indicators. We used suitable information from the corresponding questionnaire. Women questionnaire used for the place of delivery, mother's functional difficulty, mother's education, mother's age at birth, mother's life satisfaction, childbirth weight, and antenatal care visit. The questionnaire related to children aged 2–4 years was used to get information about childhood disability, birth order, and SDG related indicators like stunting and underweight. From the household questionnaire, we got information about the place of residence.

Different child, family and community level data are reported as percentage. Simple, bivariate and multivariable ordinal logistic regression analyses are used for this study. Here bivariate analysis means simple cross-tabulation (chi-square) analysis, which was used to measure the association between outcome variable and selected independent variables. The advantage of using bivariate analysis (Chi-square test) is that it provides information, whether there is a significant relationship between the outcome and independent variable. A Chi-square test is

applied with a 5% level of significance. Finally, multivariate analysis is performed using ordinal logistic regression to identify the child, family and community level factors for childhood disability. Here the outcome variable is ordinal with three categories: 0 (no disability), 1 (at least one disability) and 2 (more disabilities). Since the outcome variable is ordinal (i.e., classified according to their order of magnitude), we have used the ordinal logistic regression model to identify the significant factors. The main advantage of using ordinal regression model is that we will be able to understand which independent variable have a significant effect on the dependent variable and how well our ordinal regression model predicts the dependent variable. To perform ordinal logistic regression analysis, we recoded the significant covariate as 0 and 1. It should be mentioned that SPSS automatically takes the last category as the reference category for ordinal regression. Before running the ordinal logistic regression, we have checked whether any of the cell frequency is empty or extremely small through bivariate analysis. Although some variables showed statistically significant association in the bivariate analysis, we did not include them in the final model because of empty cells. Moreover, in the final model, we included all significant covariates and some related key variables. We estimated odds ratio (OR) and 95% confidence intervals (CI). Any p values < 0.05 were considered as statistically significant. For the overall model, the likelihood ratio test was performed. In this logistic regression analysis, multicollinearity was checked by examining the standard errors for the regression coefficient. It was considered as multicollinearity among independent variables if the standard error was greater than 2.0 [21].

### Ethical considerations

Since the data for the present study came from secondary sources, ethical approval was not required for this study. However, the survey protocol was approved by the technical committee of the Government of Bangladesh lead by Bangladesh Bureau of Statistics (BBS). The protocol included a Protection Protocol which outlines the potential risks during the life cycle of the survey and management strategies to mitigate these. Moreover, verbal consent was obtained for each respondent participating. For children aged 15–17 years who were individually interviewed, adult consent was obtained in advance of the child's assent. The interviewers mentioned that participation in the survey was voluntary and the information will be kept confidential and anonymous. Respondents had also the right to refuse answering all or particular questions, as well as to stop the interview at any time.

## Results

A total of 14,072 children under five years of age were considered for the study. Among them, 392 (2.8%) children of under five years of age, had any kind of functional difficulty. Among them 1.2% had to learn difficulty, 1.1% had difficulty in controlling behaviour, 0.7% communication, 0.5% playing, 0.4% walking, 0.3% fine motor and 0.2% had seeing and hearing difficulty. Functional difficulty in the individual domain for under-five children in Bangladesh was presented in Fig 4.

Functional difficulty in only one domain is defined as having at least one disability. Similarly, having functional difficulty with more than one domain is defined as more disabilities. The absence of any functional difficulty is categorized as no disability. As of MICS 2019 data shows that, 13,680 (97.2%) have no functional difficulty, 285 (2.0%) children have functional difficulty in at least one domain and 107 (0.8%) children have functional difficulty in more than one domain, according to mothers or primary caregivers (see Fig 5).

Table 1 presents different variables and factors associated with childhood disability (three categories). The χ2 analysis was used to see whether two variables were independent or not.

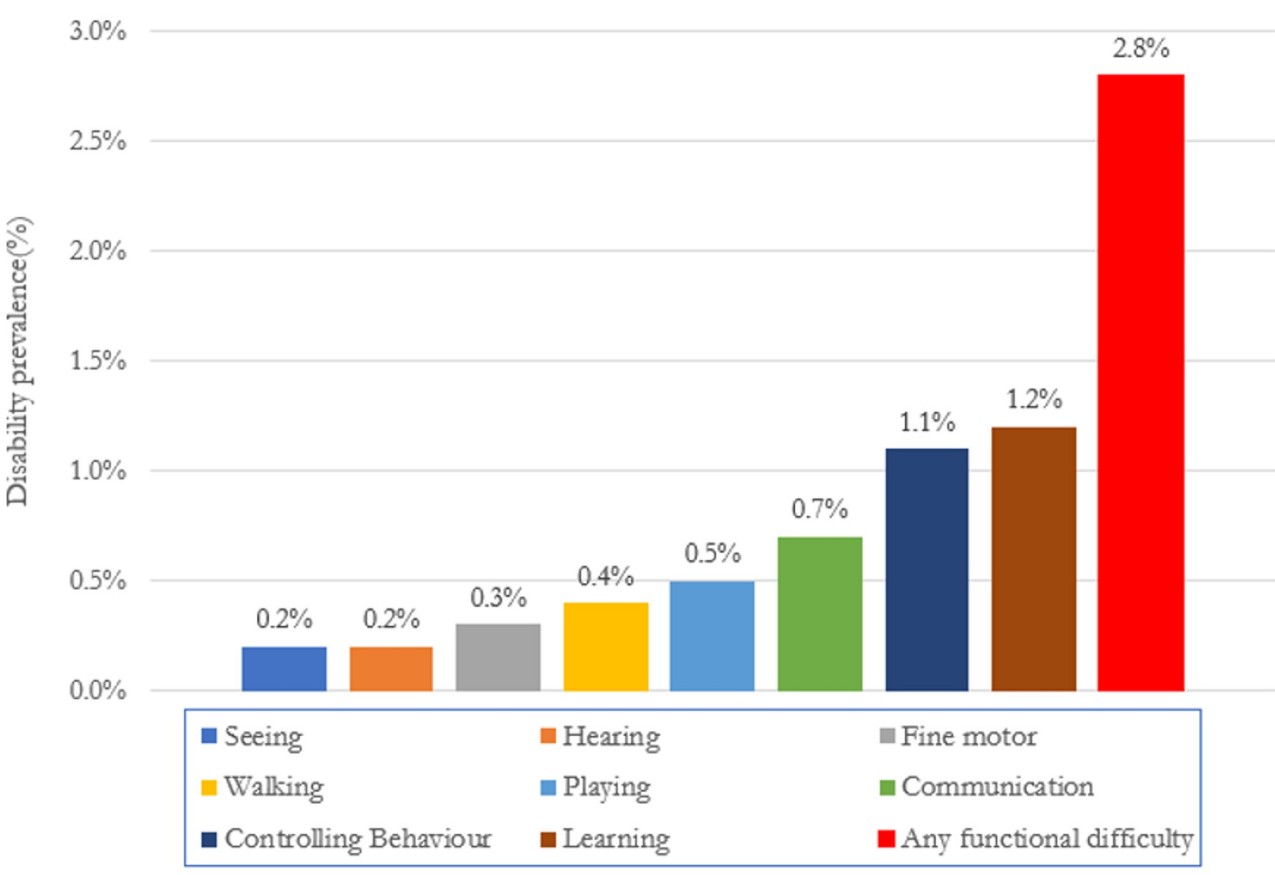

**Fig 4. Functional difficulty in the individual domain (Source: [25]).**

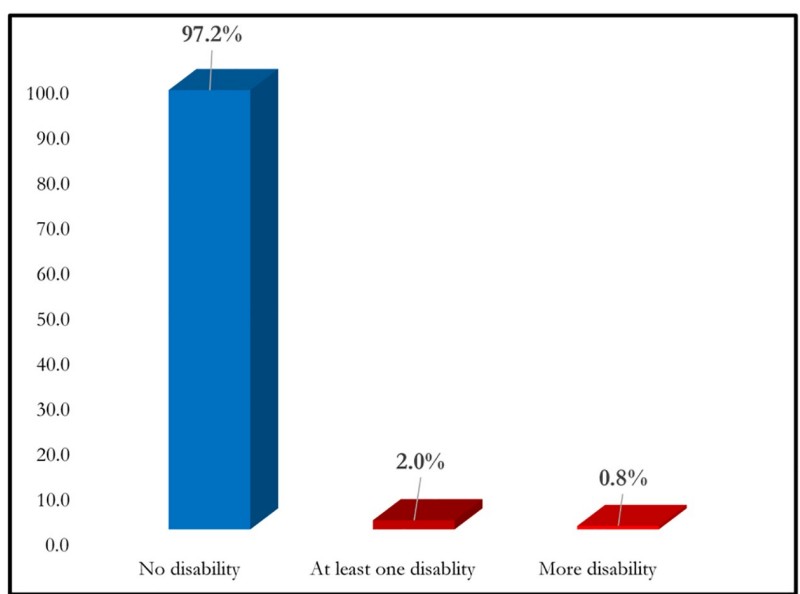

**Fig 5. Distribution of early childhood disability in Bangladesh.**

**Table 1. Prevalence of childhood disability among children aged 2 to 4 years in Bangladesh (N = 14,072).**

| Variables | | Percentage of children under age 5 | Percentage of any functional difficulty under age 5 | P value |
|---|---|---|---|---|
| **Child level variables** | | | | |
| Age | 2 | 32.8% | 3.0% | 0.682 |
| | 3 | 34.3% | 2.8% | |
| | 4 | 32.9% | 2.6% | |
| Sex | Male | 52.0% | 3.2% | <0.05 |
| | Female | 48.0% | 2.3% | |
| Early childhood education attendance | Attending | 18.9% | 1.8% | <0.05 |
| | Not attending | 81.1% | 2.9% | |
| Measures Underweight -2 SD | Not underweight | 75.2% | 2.4% | <0.01 |
| | Underweight | 24.8% | 3.4% | |
| Measure Stunt- 2 SD | Not stunted | 69.6% | 2.2% | <0.01 |
| | Stunted | 30.4% | 3.5% | |
| Birth order | First born | 83.0% | 2.7% | <0.01 |
| | Second born | 16.6% | 3.0% | |
| | Third up to last | 0.4% | 8.5% | |
| **Family level variables** | | | | |
| Mother's Age | 15–19 | 3.7% | 2.6% | <0.05 |
| | 20–24 | 26.4% | 2.4% | |
| | 25–29 | 29.5% | 2.5% | |
| | 30–34 | 23.6% | 2.3% | |
| | 35–39 | 11.8% | 3.7% | |
| | 40–44 | 3.5% | 3.3% | |
| | 45–49 | 1.4% | 6.8% | |
| Mother's education | Pre-primary or none | 12.3% | 4.1% | <0.05 |
| | Primary | 24.2% | 2.8% | |
| | Secondary | 48.6% | 2.6% | |
| | Higher secondary+ | 14.9% | 2.1% | |
| Mother's functional difficulties | Has functional difficulty | 1.6% | 10.4% | <0.01 |
| | Has no functional difficulty | 96.5% | 2.6% | |
| | No information | 1.9% | 4.8% | |
| Mother's age at birth | <20 | 51.4% | 2.4% | 0.063 |
| | 20–34 | 48.3% | 3.1% | |
| | 35+ | 0.3% | 7.5% | |
| Estimation of overall happiness | Very happy | 25.6% | 2.1% | <0.01 |
| | Somewhat happy | 60.1% | 2.6% | |
| | Neither happy or unhappy | 10.6% | 3.1% | |
| | Somewhat unhappy | 2.0% | 4.4% | |
| | Very unhappy | 1.7% | 8.4% | |
| **Community level variables** | | | | |
| Area | Urban | 21.0% | 3.3% | 0.161 |
| | Rural | 79.0% | 2.7% | |

*(Continued)*

**Table 1.** (Continued)

| Variables | | Percentage of children under age 5 | Percentage of any functional difficulty under age 5 | P value |
|---|---|---|---|---|
| Division | Barishal | 5.7% | 8.5% | <0.01 |
| | Chattogram | 22.0% | 1.3% | |
| | Dhaka | 23.6% | 4.1% | |
| | Khulna | 10.4% | 1.5% | |
| | Mymenshing | 7.4% | 5.8% | |
| | Rajshahi | 12.1% | 1.5% | |
| | Rangpur | 10.7% | 1.7% | |
| | Sylhet | 8.1% | 1.3% | |
| Antenatal care visit | 1–4 visits to any provider | 69.7% | 3.6% | 0.545 |
| | 5 or more visits to any provider | 30.3% | 6.7% | |
| Delivery at home | Yes | 44.6% | 6.3% | 0.282 |
| | No | 55.4% | 5.0% | |
| Ever used internet | Yes | 13.4% | 1.4% | <0.05 |
| | No | 86.6% | 2.9% | |
| Mobile phone owner | Yes | 79.7% | 2.8% | <0.001 |
| | No | 20.3% | 2.6% | |

All independent variables with corresponding p values (p values <0.05) were reported in Table 1. Prevalence of early childhood (aged less than 5 years) disability was found 2.8% at the time of the survey. The prevalence of disability was higher among children aged 2 years (3.0%), male children (3.2%), children who were not attending early childhood education (2.9%). Early childhood disability was significantly higher among underweight (3.4%) and stunt children (3.5%). The prevalence of disability (8.5%) was found to be higher among children in the 3-plus birth order.

Similarly, a significantly higher prevalence of early childhood disability was found among those children whose mothers age was 45 to 49 years age (6.8%) and pre-primary or uneducated mother's children (4.1%). Mothers' functional difficulty, mothers' overall happiness was also significantly associated with early childhood disability. Urban children were found to be more disabled than rural (3.3%), but the difference is not significant. Children of those mothers who visited antenatal care 5 or more times and had delivered their babies at home, early childhood disability was slightly higher among their children. Similarly, children of those families who had no access to internet and who owned a mobile phone were significantly associated with early childhood disability. Early childhood disability was also significantly higher among the Barishal division. A divisional variation on prevalence is presented in Fig 6.

The ordinal regression model for measuring the risk factors for early childhood disability among the study population was presented in Table 2. The model was fitting well. The statistically significant loglikelihood ratio test ($\chi2 = 49.925$; $p<0.001$) indicates that the model gives a better prediction for the outcome variables. The Pearson Chi-square test ($\chi2 = 1080.449$; $p = .129$) indicates that the explanatory variables were independent of each other. Additionally, the test of parallel lines indicates that the proportional odds assumption was not violated.

The ordinal regression model demonstrated that five significant variables were associated with early childhood disability in Bangladesh. Early childhood disability was higher among those children who were not attending early childhood education (OR = 0.65; 95% CI = 0.13–1.17; $p = 0.01$) than those who attained early childhood education. If the mothers were

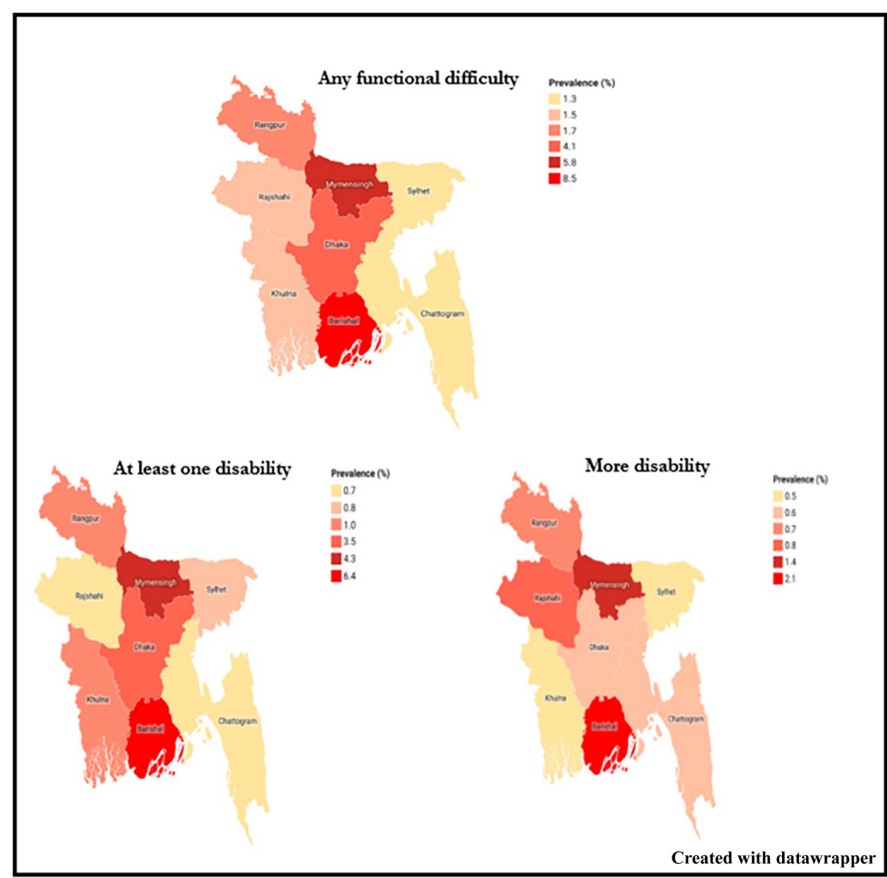

**Fig 6. Divisional variation in the prevalence of different types of disability (any functional difficulty, at least one disability to more disability) among under-five children in Bangladesh.**

disabled, then the corresponding children were more likely to be more disabled (OR = 1.23; 95% CI = 0.58–1.88; $p < 0.00$) compared with those whose mothers had no disability. Children with unhappy mothers were more likely to be more disabled (OR = 0.85; 95% CI = 0.30–1.39; $p = 0.00$) as compared to children with happy mothers. Children whose family had no access to internet connection and owned mobile phone were likely to be more disabled (OR = 0.68; 95% CI = 0.06–1.29; $p = 0.03$) and (OR = 0.52; 95% CI = 0.09–0.95; $p = 0.02$), respectively. However, no significant result was found for sex of the children, underweight, stunt, mother's education, and mothers age at birth.

## Discussion

The prevalence of early childhood (under age 5) disability rate was 2.8% in Bangladesh, where 2.0% had at least one disability and 0.8% had more than one disability. The disability also varied from district to district. This age specific disability rate is consistent with the results of another research [26]. In MICS [25], early childhood disability was measured according to mothers or primary caregivers report of children having any kind of functional difficulty in eight functional domains. Although no single definition is appropriate to measure the disability, the responses by survey respondents inevitably can shape the definition and understand of disability. Mothers' willingness to identify their children as disabled may vary according to

**Table 2. Ordinal logistic regression model for factors associated early childhood disability.**

| Variable | OR | 95% CI | P value |
|---|---|---|---|
| **Sex (Female as reference)** | | | |
| Male | **-0.22** | **- 0.53–0.09** | **0.17** |
| **Attending early childhood education (attending as reference)** | | | |
| Not attending | **0.65** | **0.13–1.17** | **0.01** |
| **Child Underweight (Not underweight as reference)** | | | |
| Underweight | **-0.03** | **-0.44–0.38** | **0.89** |
| **Child stunted (Not stunted as reference category)** | | | |
| Stunted | **0.27** | **-0.11–0 .66** | **0.17** |
| **Mother age (Less than 35 as reference)** | | | |
| 35+ | **0.24** | **-0.13–0.62** | **0.20** |
| **Mother education (Primary and above as reference)** | | | |
| Preprimary or no education | **0.15** | **-0.19–0.49** | **0.38** |
| **Mother functional difficulty (No functional difficulty as reference)** | | | |
| Has functional difficulty | **1.23** | **0.58–1.88** | **0.00** |
| **Mother age at birth (Less than 20 as reference)** | | | |
| 20+ | **0.15** | **-0.17–0.47** | **0.37** |
| **Mother overall happiness (happy as reference)** | | | |
| Unhappy | **0.85** | **0.30–1.39** | **0.00** |
| **Internet accessed (Yes as reference)** | | | |
| No | **0.68** | **0.06–1.29** | **0.03** |
| **Mobile phone user (No as reference)** | | | |
| Yes | **0.52** | **0.09–0.95** | **0.02** |

their perception of their children faces any difficulties, and the implication of defining their child as disable, as they understand them [26]. The rate of disability may vary according to the techniques and purpose of measuring disability [4]. So further research is needed to understand the valid concept and measure of childhood disability as outlines by researchers [2]. In 2016, the Global Burden of Diseases found that early childhood developmental disability was high in South Asian countries [1].

Although child mortality is declining globally, the burden of childhood disability is not decreasing. This is particularly true for those children who were born prematurely. These children are making a big contribution to the burden of disability [1]. Moreover, children in developing countries have poorer physical health, which is also a risk factors for childhood disability [3]. However, this study used ordinal logistic regression analysis to find out the associated factors for functional difficulty at least one domain to more. These findings have a strong association among child level, family level and community level factors for disability. Child-level factors including early childhood education, family level factor including mothers have functional difficulty and mother's overall life satisfaction and community level factors including accessed to internet and own a mobile phone. Different studies show that mother's functional difficulty, early education attendance, mothers life satisfaction and accessed internet are important factors for occurring early childhood disability [2, 16, 27–29].

The disability is high among children who do not attain the early educational program than those who attain these programs. To contribute to the society, disable children must attain early education program. By early educational program, they can grow up with confidence which makes them competent learner and communicators, healthy in mind with good body function [16, 30]. Most of the disabled children in developing countries including Bangladesh

are unable to attain early education programs (through formal, nonformal, or informal) opportunities due to social stigma and ignorance [31]. Regardless of whatever quality of education accessed of disabled children, 40% of disabled children come from advantaged families and can afford to pay their education cost. Although, Bangladesh government has taken several initiatives to provide free educational facilities for all disabled children, few non-government organization-imposed tuition fees for the early education program of disable children [31]. By addressing the negative attitude including institutional discrimination toward disabling children, mainstreaming can be increased. By ensuring quality education in school can help every disabled child to reach their full potential [31, 32]. That will ultimately focus only on the children's ability may ultimately reduce the burden of disability. So, the government needs to ensure early education for all disabled children and also make awareness of disabled children's parents [31, 32].

Mothers having functional difficulties were highly significantly associated with early childhood disability compared to those whose mothers did not have any functional difficulty. Mothers' functional difficulty results from illiteracy, unemployment, consanguineous marriage, and multiparity. These influences negative child health outcomes such as disability [2]. Many genetical studies support these results and they reported if a mother had functional difficulty their children were two times more likely to be disabled than those whose mothers didn't have any difficulty [29, 33–35]. Moreover, children with disable parents could experience some negligence because of their mother's inability to take care of them; these social contexts may create childhood disability to some extent. It is urgent to collect data for family health history and free screening for all newborn babies to detect disability and for early intervention [29, 33–35].

Our study shows that mothers' overall life satisfaction has a significant impact on early childhood disability. If a mother was dissatisfied, their children were more likely to be disabled than those whose mothers were satisfied in their life. Genetical studies support that mother's depression can easily transmit to their children and that may cause behavioral problems among children [29]. Maternal unhappiness has an impact on maternal childcaring behavior. The unhappy mother may be less involved and more negative about their child nurturing. That may reduce the children's cognitive function and create a lot of behavioral problems [36, 37]. The happiness of the mother can bring good mental health among children and protect them from disability especially on the poor behavioral problem. Maternal social support can prevent early childhood disability [16, 38]. A program that could identify the young children's mother with poor mental health and an unhappy relationship with a partner is needed to give those families beneficial support. These actions may help them to decrease parental stress, focus on the value of communication, and strengthen the mother's ability. It also helped the mother to realize that stress caused by raising a child with a disability is a normal part of their life [36].

For parents of children who did not access to the internet, early childhood disability is significantly high among their children in comparison to those parents who accessed the internet. From this result, it was indicated that parents can get various information regarding childhood diseases, deliveries, breastfeeding, and disability. So, they can aware of children's problems and can take initiatives to prevent childhood disability. Similar results are found in the different studies, which find the link between parents not using the internet and childhood disability [26, 27]. A study conducted by Islam et al., (2016) [22] suggested that by improving the social network, disable children including their families can be benefitted through sharing personal matters, looking for help, and consulting in the time of important decision making. Parents' using a mobile phone has also a significant impact on early childhood disability. A recent time study indicates that most of the parents gave their children mobile phones while feeding and

their children were using a mobile phone before age of 2 years. Childhood neurodevelopmental disability can occur due to early and frequent mobile media habits among young children, lack of awareness about the bad impact of mobile [28]. A clear medical guideline is urgently needed regarding mobile phone usage among young children with neurodevelopmental disability.

Several studies find different significant factors for early childhood disability. In Bangladesh, malnutrition including stunting, underweight, low birth weight, deficiency of vitamin A, anemia, and iodine deficiency is one of the major factors causing early childhood disability in Bangladesh [10]. Maulik and Darmstadt (2007) [2] found that the factors for childhood disability in low- and middle-income countries are genetic diseases, the mother with older age, mother illiteracy, children delivery at home, low birth weight and lack of antenatal care. Environmental factors such as product and technology, chemicals, human-made changes to the environment, support and relationships, attitudes, services, systems, and policies have an impact on disability and all components of functioning [15]. Geographical features, as well as policy, have an impact on disability. Poverty is also a significant risk factor for childhood disability [4]. But, in the present study, sex, stunted, underweight, mother age, and mother education failed to show any significant impact on early childhood disability.

Although Bangladesh has made notable progress in preventing early childhood disability, the progress is not consistent due to poverty, illiteracy, insufficient maternal and neonatal health care. Disable children receive inadequate attention in the health sector, insufficient disability education and awareness, disability rights have not always been addressed in national policies, many children's homes are located in areas where rehabilitation services are inaccessible [10]. Bangladesh government has taken several initiatives to improve the nutritional status, new educational policies for disabled children, inclusive early childhood education that may contribute to reducing the burden of early childhood disability in Bangladesh. According to the United Nations Conventions on the Rights of the Child and Rights of Persons with Disabilities, disable children including their families can enjoy full and equal rights [10]. Government can take initiatives to provide necessary assistance to disable children so that they can achieve full potential. The negative consequences of childhood disability can be addressed by identifying local factors for disability as well as developing effective strategies to deal with them [10]. Different types of strategies include sponsoring different effective health education programs that highlight the role of genetic factors for causing disability, effective community-based maternal and child health care services, improving mother awareness about different types of available services for disabilities, and adequate nutritional supplementation programs [2].

The study is one of the leading studies to find out the factors associated with early childhood disability. The analysis is based on the nationally representative and large dataset based on MICS 2019 [25], where data were collected through a face-to-face interview. As a consequence, the results can be generalized nationwide. We have calculated the national burden of under-five functional difficulty in Bangladesh. Another strength of this study is to find out the child level, family level, and community factors that are contributing to the early childhood disability in Bangladesh. However, our research also suffers from some limitations. The analysis of this study is based on cross-sectional data, so we cannot draw any conclusion about the causal relationship between the selected factors and early childhood disability. Moreover, the study used only parents/caregivers' reports to measure functional difficulty, clinical diagnosis was not included. This may cause some biases in the research findings. Despite of these limitations, this research provides nationally representative evidence-based information. This research result can help understand the factors linked to early childhood disability in Bangladesh. This study results will be helpful for policymakers and others stakeholders to formulate

early interventions and feasible solutions for early childhood disability. It may also suggest the appropriate programs, which may ultimately reduce the overall burden of disability.

## Conclusion

Early childhood disability is a significant health issue in both low- and middle-income countries because it negatively impacts their quality of life and productivity. The study revealed that 2.8% of children under five years of age are suffering from any kind of functional difficulty. The significant factors for early childhood disability were early childhood school unattendance, mother functional difficulty, less overall life satisfaction of mothers, no family access to internet and ownership of mobile phone. Based on our results, a comprehensive community-based initiative should be developed and implemented throughout the country to resolve the early childhood disability issue. Early school attendance for disabled children could be improved by removing the barrier of sociocultural attitude and practices, which will help disable children to full acceptance, participation and improve learning ability. Improving educational attainment of disable children will make them independent. Moreover, special education should be provided for those disabled children who are unable to attain mainstream education. Large-scale genomic data for parents (physical or mental difficulty) can be used to identify the genetic causes for early childhood disability, which may help for early diagnosis. Identifying a mother's depression, providing support and treatment during the time of antenatal could be helped to reduce stress as well as avoid long-term negative outcomes in their children. Disability-related information must be made accessible in different media (newspaper, television, and radio). Disable children's parents often engaged their children with a mobile device when they involved themselves in different activities. A strong clinical guideline is needed for using the mobile device of under-five children.

The government and others policymakers should concentrate on these interventions that could be helped to reduce the overall burden of early childhood disability. That also includes, effecting planning should be taken to support and services for the disabled children and their families. However, the present findings recommended that further large-scale study is needed to develop a tool to measure parents' genetic disability and how environmental factors affect early childhood disability.

## Supporting information

**S1 Table. Table of the outcome and independent variables clearly describing their potential nature and values and codes.**
(DOCX)

**S2 Table. The exact items assessing the different types of disabilities.**
(DOCX)

## Acknowledgments

Authors wish to acknowledge UNICEF, MICS for providing access to the data set.

## Author Contributions

**Conceptualization:** Shilpi Rani Saha.

**Writing – original draft:** Shilpi Rani Saha.

**Writing – review & editing:** Md. Mobarak Hossain Khan.

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
