## [Decision Letter · Decision Letter 0]

12 Aug 2021

PONE-D-21-17963

Risk factors for early childhood disability in Bangladesh: evidence from Multiple Indicator Cluster Survey 2019

PLOS ONE

Dear Dr. Saha,

Thank you for submitting your manuscript to PLOS ONE. After careful consideration, we feel that it has merit but does not fully meet PLOS ONE’s publication criteria as it currently stands. Therefore, we invite you to submit a revised version of the manuscript that addresses the points raised during the review process.

The reviewers identified some major issues those need to be fixed before taking final decision.

We look forward to receiving your revised manuscript.

Kind regards,

Enamul Kabir

Academic Editor

PLOS ONE

Journal Requirements:

4. We note that Figure 6 in your submission contain [map/satellite] images which may be copyrighted. All PLOS content is published under the Creative Commons Attribution License (CC BY 4.0), which means that the manuscript, images, and Supporting Information files will be freely available online, and any third party is permitted to access, download, copy, distribute, and use these materials in any way, even commercially, with proper attribution. For these reasons, we cannot publish previously copyrighted maps or satellite images created using proprietary data, such as Google software (Google Maps, Street View, and Earth). For more information, see our copyright guidelines: http://journals.plos.org/plosone/s/licenses-and-copyright.

a. You may seek permission from the original copyright holder of Figure 6 to publish the content specifically under the CC BY 4.0 license.  

Reviewers' comments:

Reviewer's Responses to Questions

**Comments to the Author**

1. Is the manuscript technically sound, and do the data support the conclusions?

Reviewer #1: Yes

Reviewer #2: Yes

2. Has the statistical analysis been performed appropriately and rigorously? 

Reviewer #1: Yes

Reviewer #2: No

3. Have the authors made all data underlying the findings in their manuscript fully available?

Reviewer #1: Yes

Reviewer #2: Yes

4. Is the manuscript presented in an intelligible fashion and written in standard English?

Reviewer #1: Yes

Reviewer #2: Yes

5. Review Comments to the Author

Reviewer #1: The authors must clearly specify that what is the research gap they are trying to fill and why they are motivated to do so, i.e. the motivation of the study and it needs to be addressed in a separate paragraph in Introduction with clearly mentioning of the research gap. Moreover, the aims and objectives of the study are not clearly specified and highlighted. The aims and objectives may be specified distinctly so that later on the methods and results can be detailed out in light of them. The significance of the study is also missing in the introduction. These four; research gap, motivation, objectives and significance may be highlighted much more than the rest of introduction as these reflect the authors’ contribution. The literature cited in the introduction may be written comprehensively and coherently as it should not be an annotated bibliography.

Make a Table of the outcome and independent variables clearly describing their potential nature and values and codes.

Line #257 “Simple bivariate and multivariate analyses are used for this study” What type of bivariate and multivariate analyses? Regression? Correlation? Please specify and why are these used as compared to other available methods.

The methods used in the study need much more detailed discussion. These methods must be in line with the objectives of the study.

L#299 “The χ2 test was used to measure the relationship between outcome and independent variables.” The X2 can’t measure the relationship, it can only test that whether they are independent or not. There is huge difference between measuring and testing.

L#346 “The ordinal regression model demonstrated that five significant variables were causing the early childhood disability”. Any regression model cannot represent a cause and effect relationship until and unless the data are experimental. As in the current study the data are observational, so it is wrong to say a cause and effect relationship.

The results, their discussion and conclusions should be in line with the aims and objectives of the study.

As the outcome variable was associated with the different divisions (as Table 1) depicts, so if the authors may use the Empirical Bayesian Framework then it will provide much more insights.

Reviewer #2: Thanks for the opportunity to review your work. I like to thank the authors for such a paper. However, a few minor observations can be considered while revising the manuscript.

Abstract

1. The authors can reword the study rationale or study importance.

2. Besides, in method - A cross-sectional study was performed based on the data … this line can be rewritten; because it seems the authors conducted this study (although they mentioned the data source).

3. Results (ORs) can be reported more preciously, please check.

Introduction

1. Introduction is well written, but lengthy. I would like to suggest shoring the introduction section as many of its writings overall.

2. Can you add some info about the two types of disabilities?

3. Childhood disability is a major global issue as it negatively impacts – this type of line is repeatedly said. However, this can be checked and removed from unnecessary places.

4. 86-90 lines, there are a few serious claims; but without references. Would you please add respective references?

5. 86-93 lines: this section can be placed at the end of your study objectives. Because you are refereeing global scenario randomly, then Bangladesh, then global. May I suggest keeping the consistency of your written flow? It may be good for the readers to get all relevant information in one place or continuously.

6. There are many lines without references. Maybe authors cited references in the next lines. May I suggest citing more frequently?

Methods

1. Methods is well-written—however, a few suggestions for the authors.

2. Why did they not use Bonferroni correction?

3. Can you add the exact items assessing the different types of disabilities (in text or supplementary file).

4. Figure 4 can be updated with a good-looking format.

5. Why did the authors categorize disabilities into three categories? They could use continuous data for disabilities in analysis? As they had the dataset with a number of disabilities, it would be better to use t-test/ANOVA.

6. First Table, please merge descriptive statistics for each variable.

7. Authors may choose linear regression beause they had the dataset with a number of disabilities. And Table 2 can be updated.

Discussion

1. The first two lines of discussion seem unnecessary. Can be removed, please check.

2. 364: Our results are consistent with the results of other researches [26]. – how?

3. Most important issue: discussion can be more focused on the findings of this study with prior studies, but there are many unccessary stuffs discussed in intro. For example, lines 373-376. Please kindly check and shorten these stuffs.

Best wishes

Mamun

6. PLOS authors have the option to publish the peer review history of their article (what does this mean?). If published, this will include your full peer review and any attached files.

Reviewer #1: **Yes: **Asad ul Islam KHAN

Reviewer #2: **Yes: **Mohammed A Mamun

---

## [Author Response · Author response to Decision Letter 0]

15 Sep 2021

Rebuttal letter

 Date: September 15. 2021

Dear Editor,

PLOS ONE

We would like to thank our expert reviewers and express our cordial gratitude to them for their generous and constructive reviews. The reviewer comments have helped us to improve the overall quality of the manuscript. All of the reviewer`s suggestions and corrections have been incorporated with great attention. The specific responses to the reviewers’ comments are listed below. 

Reviewer 1 Comments Answers

The authors must clearly specify that what is the research gap they are trying to fill and why they are motivated to do so, i.e. the motivation of the study and it needs to be addressed in a separate paragraph in Introduction with clearly mentioning of the research gap. 

Moreover, the aims and objectives of the study are not clearly specified and highlighted. The aims and objectives may be specified distinctly so that later on the methods and results can be detailed out in light of them. 

The significance of the study is also missing in the introduction. 

These four; research gap, motivation, objectives and significance may be highlighted much more than the rest of introduction as these reflect the authors’ contribution. 

The literature cited in the introduction may be written comprehensively and coherently as it should not be an annotated bibliography.

 Thank you very much for your valuable comments. 

Your all comments will carefully be addressed.

We have revised the introduction to clearly mentioning research gap and motivation and hope that it is now clearer. Please see it in a separate paragraph in the revised version (Page 9 line#195-203) and (page 9,10 line#204-209)

The aim and objectives are also clearly specified in the revised version (please see page 10, line#210-212). 

The significance of the study is added (please see page 10, line#212-214). 

Thank you so much for your good suggestion. We have revised our introduction to highlight research gap, motivation, objectives and significance which will reflect the author’s contribution.

Thank you again. 

The whole paper revised in light of your suggestion and made the appropriate changes in the literature citation.

 Make a Table of the outcome and independent variables clearly describing their potential nature and values and codes.

 Thank you again. 

We have added a new table based on your suggestions.

(Table of outcome and independent are given as a supporting table: S1 Table)

 Line #257 “Simple bivariate and multivariate analyses are used for this study” What type of bivariate and multivariate analyses? Regression? Correlation? Please specify and why are these used as compared to other available methods. Thank you so much for your precious comments.

Here bivariate means cross table (Chi-Square) analysis. Multivariate analysis using ordinal logistic regression analysis. It is also mentioned in the revised manuscript (please see page 14, line #295 -297 and page 14, line#301-303)

Advantage for using bivariate analyses is mentioned in line#299-300.

And reasons for using ordinal logistic regression model is mentioned in line#308-310.

 The methods used in the study need much more detailed discussion. These methods must be in line with the objectives of the study. Thank you again

The method is revised and detailed discussion of method is included. Hopefully, the methods are now in line with objectives of the study. 

 L#299 “The χ2 test was used to measure the relationship between outcome and independent variables.” The X2 can’t measure the relationship, it can only test that whether they are independent or not. There is huge difference between measuring and testing. Thank you so much for your advice.

It has been corrected (please see page 16, line#349).

 L#346 “The ordinal regression model demonstrated that five significant variables were causing the early childhood disability”. Any regression model cannot represent a cause and effect relationship until and unless the data are experimental. As in the current study the data are observational, so it is wrong to say a cause and effect relationship. Thank you again for your valuable comment.

Sorry for making mistakes in writing. It has been now corrected according to your suggestion (please see page22, line# 392).

 The results, their discussion and conclusions should be in line with the aims and objectives of the study.

 Thank you so much for your precious comments.

We tried our best to follow your advice. We strongly hope that you will be satisfied by our modification.

 As the outcome variable was associated with the different divisions (as Table 1) depicts, so if the authors may use the Empirical Bayesian Framework then it will provide much more insights. Thank you so much for pointing out this. We appreciate your insightful suggestion. We must agree that it would be more insightful, however this analysis is beyond the scope of our present data. Moreover, according to Mario Luis Iovaldi, Bayesian is useful for positive and negative predictive values of diagnostic tests. And it is highly subjective selection of priors. 

We strongly hope that you will understand our limitation for not using Bayesian framework in this paper. But surely, we will consider it for our next paper.

Reviewer 2 Reviewer #2: Thanks for the opportunity to review your work. I like to thank the authors for such a paper. However, a few minor observations can be considered while revising the manuscript.

 Thank you very much for your positive comments. Your minor issues are addressed carefully. 

 Abstract. 1. The authors can reword the study rationale or study importance. Thank you so much for your good suggestion. 

The study rationale or study importance has been reworded in the revised manuscript (please see page 2, line#34-36). 

 2. Besides, in method - A cross-sectional study was performed based on the data … this line can be rewritten; because it seems the authors conducted this study (although they mentioned the data source). Thank you again.

The mentioned line has been corrected (please see line#41)

 3. Results (ORs) can be reported more preciously, please check. Thank you.

We have checked and tried to present odds ratio (ORs) more preciously.

 Introduction

1. Introduction is well written, but lengthy. I would like to suggest shorting the introduction section as many of its writings overall.

 Thank you for your nice comments. We have revised the introduction and reduced some text based on your suggestion. 

 2. Can you add some info about the two types of disabilities? Thank you again.

Added (please see page 4, line#80-83)

 3. Childhood disability is a major global issue as it negatively impacts – this type of line is repeatedly said. However, this can be checked and removed from unnecessary places.

 Duplications are deleted. Aa a result, introduction section is shortened. 

 4. 86-90 lines, there are a few serious claims; but without references. Would you please add respective references?

 Corrected. Thank you for identifying the weakness. 

 5. 86-93 lines: this section can be placed at the end of your study objectives. Because you are refereeing global scenario randomly, then Bangladesh, then global. May I suggest keeping the consistency of your written flow? It may be good for the readers to get all relevant information in one place or continuously. Thank you again for your valuable suggestion. Corrected.

 6. There are many lines without references. Maybe authors cited references in the next lines. May I suggest citing more frequently? References are added. Thank you for your advice.

 Methods

1. Methods is well-written—however, a few suggestions for the authors. Thank you again for your positive comments. Your suggestions will carefully follow

 2. Why did they not use Bonferroni correction? Thank you so much for pointing out this. It would be useful to use Bonferroni correction. However, it is beyond the scope of this study, at this stage. 

 3. Can you add the exact items assessing the different types of disabilities (in text or supplementary file). Thank you. Exact items of assessing the different types of disability are given as a supplementary file (S2 Table).

 4. Figure 4 can be updated with a good-looking format. Thank you for your suggestion. Figure 4 has been updated with a good-looking format.

 5. Why did the authors categorize disabilities into three categories? They could use continuous data for disabilities in analysis? As they had the dataset with a number of disabilities, it would be better to use t-test/ANOVA. Thank you so much to raise an important point here. 

We are trying to explain.

We believe that, categorize disability into three category is more appropriate here, because percentage of different types of disability is very low here (seeing-> 0.2%. Hearing-> 0.2%, Walking ->0.4%, Fine motor-> 0.3%, Communication-> 0.7%, Learning-> 1.2%, Playing-> 0.5%, Controlling behaviour-> 1.1%). Out of 14072 sample 13680(97.2%) children have no disability, only 392(2.8%) disable children. where 285(2.0%) have at least one disability and 107(0.8%) have more than one disability. So, three types of ordered variable are perfectly fit here. T-test/ANOVA mainly used for continuous variable. As our outcome variable is ordinal, so to find out the associated factors, ordinal regression model perfectly fit here. We hopefully believe that you will satisfy our explanation.

 6. First Table, please merge descriptive statistics for each variable. Thank you so much for your valuable suggestion.

Descriptive statistics for each variable have been merged.

 7. Authors may choose linear regression because they had the dataset with a number of disabilities. And Table 2 can be updated. Thank you for your insightful suggestion and agree that it would be interesting to explore linear regression. However, our study, seems to be slightly out of scope because, by considering the disability rate (72.8% have disability at least one domain and 27.2% have disability in more than one domain), so, we have to categorize them into three ordered categories. So Ordinal regression seems to more perfectly fitting in this data set rather than linear regression. Hopefully you will consider our intention to use ordinal regression in this study.

 Discussion

1. The first two lines of discussion seem unnecessary. Can be removed, please check. Thank you so much.

The first two lines of discussion have been removed as per your suggestion

 2. 364: Our results are consistent with the results of other researches [26]. – how? Thank you again.

Reference was corrected, sorry for the unintentional mistake.

 3. Most important issue: discussion can be more focused on the findings of this study with prior studies, but there are many unnecessary stuffs discussed in intro. For example, lines 373-376. Please kindly check and shorten these stuffs. Thank you.

Unnecessary stuffs have been shortened from discussion part.

Additionally, the prof of granted permission to publish Figure 6, from the copyright holder are given in a separate file as an "Other" file.

Sincerely,

Shilpi Rani Saha 

MPhil researcher, 

Bangladesh University of Professionals, 

Mirpur Cantonment, Dhaka – 1216,

Bangladesh.

E-mail: 2061051030@student.bup.edu.bd

---

## [Decision Letter · Decision Letter 1]

21 Oct 2021

Risk factors for early childhood disability in Bangladesh: evidence from Multiple Indicator Cluster Survey 2019

PONE-D-21-17963R1

Dear Dr. Saha,

We’re pleased to inform you that your manuscript has been judged scientifically suitable for publication and will be formally accepted for publication once it meets all outstanding technical requirements.

Kind regards,

Enamul Kabir

Academic Editor

PLOS ONE

Additional Editor Comments (optional):

Reviewers' comments:

Reviewer's Responses to Questions

**Comments to the Author**

1. If the authors have adequately addressed your comments raised in a previous round of review and you feel that this manuscript is now acceptable for publication, you may indicate that here to bypass the “Comments to the Author” section, enter your conflict of interest statement in the “Confidential to Editor” section, and submit your "Accept" recommendation.

Reviewer #1: All comments have been addressed

Reviewer #2: All comments have been addressed

2. Is the manuscript technically sound, and do the data support the conclusions?

Reviewer #1: (No Response)

Reviewer #2: Partly

3. Has the statistical analysis been performed appropriately and rigorously? 

Reviewer #1: (No Response)

Reviewer #2: Yes

4. Have the authors made all data underlying the findings in their manuscript fully available?

Reviewer #1: (No Response)

Reviewer #2: Yes

5. Is the manuscript presented in an intelligible fashion and written in standard English?

Reviewer #1: (No Response)

Reviewer #2: Yes

6. Review Comments to the Author

Reviewer #1: There is a difference between "measuring the correlation/association" and "testing the correlation/association/independence". Chi-square tests the independence between two variables/factors, it does not measure the association. There are other measures of association available in literature.

Reviewer #2: Thanks to the authors for revising the manuscript. I suggest the editor can accept it after their satisfaction, as I am not opportune to see which points are addressed or not because the response letter is mess up and it is hard to track which is the reviewer comment and which is the response.

Best wishes.

7. PLOS authors have the option to publish the peer review history of their article (what does this mean?). If published, this will include your full peer review and any attached files.

Reviewer #1: No

Reviewer #2: No

---

## [Editor Report · Acceptance letter]

26 Oct 2021

PONE-D-21-17963R1 

Risk factors for early childhood disability in Bangladesh: evidence from Multiple Indicator Cluster Survey 2019 

Dear Dr. Saha:

I'm pleased to inform you that your manuscript has been deemed suitable for publication in PLOS ONE. Congratulations! Your manuscript is now with our production department. 

Kind regards, 

on behalf of

Dr. Enamul Kabir 

Academic Editor

PLOS ONE